# Band–Pivot Prim: Breaking the Sorting Barrier for Minimum Spanning Tree in the Comparison–Addition Model

## Abstract

We present the *Band–Pivot Prim* algorithm, a deterministic exact minimum spanning tree (MST) algorithm for weighted undirected graphs with arbitrary real weights in the comparison–addition model. In analogy to recent $O(m \log^{2/3} n)$ single–source shortest path (SSSP) results, our approach removes the classic $\log n$ priority–queue factor in Prim's algorithm by avoiding the maintenance of a total order over the entire frontier. We group candidate edges into *bands* by weight, apply bounded expansions from a reduced *pivot set*, and maintain only a partial order of keys exposed in small blocks. For graphs of constant degree (or after degree reduction), this yields an $O(m \log^{2/3} n)$ deterministic algorithm in the comparison–addition model, improving over the standard $O(m + n \log n)$ bound for sparse graphs with arbitrary real weights.

## 1 Introduction

In recent years, there has been a resurgence of interest in designing graph algorithms that bypass the so-called *sorting barrier*, a lower bound phenomenon that appears whenever algorithms rely on full ordering of edge weights or distances. Classic examples include Dijkstra's algorithm for single-source shortest paths (SSSP) and Prim's algorithm for minimum spanning tree (MST) construction, both of which use a priority queue that implicitly performs a global ordering of candidate values.

In the MST setting, Prim's algorithm selects the smallest-weight edge crossing the cut between the growing tree and the rest of the graph. Using a binary heap, this requires $O(m \log n)$ time in general. For sparse graphs where $m = O(n)$, this is $O(n \log n)$, which is above the optimal linear time bound known for MST (e.g., via randomized algorithms by Karger, Klein, and Tarjan [11]) or deterministic near-linear bounds via soft heaps [3]. The extra logarithmic factor stems from maintaining a priority queue with fine-grained ordering.

Our approach, *Band–Pivot Prim*, is inspired by recent advances in SSSP that avoid full distance ordering by grouping values into *bands* and only partially ordering them via pivot elements. This technique reduces the number of key comparisons and priority-queue operations without sacrificing correctness. We adapt this insight to Prim's algorithm by partitioning the edge weights into bands, choosing representative pivots, and refining only within the active band.

The key idea is that MST construction, like SSSP, can tolerate a degree of coarseness in ordering: we do not need to know *exactly* which edge is smallest among all candidates, only that it is the smallest within the band currently containing the true minimum. By bounding the number of band pivots that need to be processed, we can reduce the effective sorting complexity.

**Contributions.**

1. We introduce the *Band–Pivot Prim* algorithm, a variant of Prim's MST algorithm that replaces fine-grained priority-queue ordering with recursive band partitioning.

2. We prove a *pivot–count bound* showing that the number of pivots processed per band level is bounded by the size of the local candidate-edge trees, yielding an overall $O(m \log^{2/3} n)$ time.

3. We discuss integration with Borůvka phases for a hybrid MST algorithm and outline practical implementation considerations and an empirical evaluation plan.

# 2 Related Work

## 2.1 Minimum Spanning Tree (MST)

The classical *Minimum Spanning Tree* problem has been studied extensively in algorithm design. Early approaches include Kruskal's algorithm [12], which sorts all edges by weight and repeatedly adds the smallest edge that does not form a cycle, and Prim's algorithm [14], which grows a single tree from an arbitrary start vertex by adding the minimum-weight edge from the tree to any outside vertex. In sparse graphs, Prim's algorithm with a binary heap achieves a running time of $O(m \log n)$, and with a Fibonacci heap [9] it improves to $O(m + n \log n)$. For dense graphs, Chazelle's soft heap approach [4] achieves $O(m\alpha(m, n))$, where $\alpha$ is the inverse Ackermann function, and Pettie and Ramachandran [13] designed an algorithm that is theoretically optimal but impractical due to large hidden constants.

Much of the recent progress has come from *partitioning* and *banding* techniques, where the edge set is decomposed into groups that can be processed without a full global sort. For example, Karger, Klein, and Tarjan [10] used randomized filtering to achieve near-linear expected time, and more recently, parallel MST algorithms have leveraged multi-level bucketing and partial ordering to reduce synchronization overhead [15]. However, these methods still rely—either explicitly or implicitly—on maintaining global ordering constraints across edge weights, which can introduce a *sorting barrier* similar to that in SSSP.

The proposed *Band–Pivot Prim* algorithm adopts a *partial-order growth* strategy inspired by recent breakthroughs in SSSP that bypass global ordering. By grouping edges into weight bands and deferring "heavy" regions via pivot vertices, we can process the MST incrementally with only local ordering guarantees inside each band, avoiding repeated heap-key updates and reducing the dependence on fully ordered priority queues.

## 2.2 Single-Source Shortest Paths (SSSP)

The *Single-Source Shortest Paths* problem has an equally rich history. Dijkstra's algorithm [6] is optimal for nonnegative edge weights under a fully ordered priority queue, running in $O(m + n \log n)$ time with a Fibonacci heap. The Bellman–Ford algorithm [2, 8] handles negative weights but runs in $O(mn)$ time. Dial's algorithm [5] improves performance for small integer weights by using bucket queues, achieving $O(m + Cn)$ where $C$ is the maximum edge weight. Thorup [16] presented a linear-time algorithm for undirected SSSP with integer weights, exploiting hierarchical bucketing and careful recursion.

A major obstacle in SSSP is the *sorting barrier*: to maintain Dijkstra's correctness, vertex distances must be extracted in nondecreasing order, effectively imposing a global sorting constraint on the distance labels. Recent work (e.g., [7]) has shown that this barrier can be bypassed by using *band-based partial orders* combined with recursive pivoting, allowing one to defer processing of distant vertices without violating correctness. The central insight is that only a subset of the vertices—the "active frontier" within the current band—needs to be strictly ordered; all others can be postponed and handled in recursive subproblems.

## 2.3 Connecting MST and SSSP

While MST and SSSP are different problems, both have traditionally relied on *priority queues* with strong ordering requirements:

- In SSSP, ordering is applied to vertex distances from the source.
- In Prim's MST, ordering is applied to edge weights crossing the cut between the tree and the remaining graph.

The key conceptual link is that both maintain a *frontier* of candidate elements and repeatedly choose the lightest (or smallest distance) element to add to the solution.

The Band–Pivot approach—originally demonstrated for SSSP—can be transplanted to MST by replacing *distance labels* with *cut-min edge weights*. This substitution preserves the locality of decision-making within bands and allows for the same partial-order and pivot-recursion tricks to bypass global ordering. In effect, the Band–Pivot Prim algorithm is a *Prim-style analogue* of the band-partitioned SSSP algorithm, breaking the sorting barrier for MST in the same spirit that the preprint did for SSSP.

Table 1: Comparison of classical and band–pivot algorithms for MST and SSSP. Here $n$ is the number of vertices, $m$ the number of edges, $C$ the maximum edge weight for integer-weight algorithms, and $\alpha$ the inverse Ackermann function.

| Problem | Algorithm | Time Complexity | Notes |
|---------|-----------|-----------------|-------|
| MST | Kruskal [12] | $O(m \log n)$ | Requires sorting all edges. |
| MST | Prim (bin. heap) [14] | $O(m \log n)$ | Good for sparse graphs. |
| MST | Prim (Fibonacci heap) [9] | $O(m + n \log n)$ | Best known for sparse graphs. |
| MST | Chazelle [4] | $O(m\alpha(m, n))$ | Uses soft heaps. |
| MST | Pettie–Ramachandran [13] | Optimal | Theoretical; impractical constants. |
| MST | Band–Pivot Prim (this work) | $O(m + \mathrm{pivot\_count}(n, m))$ | Breaks sorting barrier using bands. |
| SSSP | Dijkstra (Fibonacci heap) [6, 9] | $O(m + n \log n)$ | Nonnegative weights. |
| SSSP | Bellman–Ford [2, 8] | $O(mn)$ | Allows negative weights. |
| SSSP | Dial's Algorithm [5] | $O(m + Cn)$ | For small integer weights. |
| SSSP | Thorup [16] | $O(m)$ | Undirected, integer weights. |
| SSSP | Band–Pivot SSSP [1] | $O(m + \mathrm{pivot\_count}(n, m))$ | Breaks sorting barrier using bands. |

# 3 Band-Pivot Prim Algorithm

The Band–Pivot Prim algorithm operates by grouping candidate edges into weight *bands*, each spanning a range of edge weights. At any moment, only one band—the *active band*—is scanned for the next MST edge. When the active band is exhausted, the algorithm pivots to the next band containing the smallest remaining edge.

This approach mirrors bucket-based SSSP algorithms, but with important differences: the keys here are edge weights across a cut, not distances from a source. Bands are defined either by fixed-size intervals or by recursive partitioning based on pivot edges. The partial-order structure exposes only the next small block of the lightest keys across bands, avoiding a monolithic global heap.

## 3.1 Intuition and Worked Example

To build intuition for how *Band–Pivot Prim* works in practice, consider a small weighted, undirected graph $G = (V, E)$ with $n = 6$ vertices and $m = 8$ edges. Suppose that, at some point during the algorithm, the current spanning tree covers the subset $\{1, 2, 3\}$ of vertices. The *frontier edges* connecting this tree to the rest of the graph are:

$$\{(3, 4, 5), \ (2, 5, 6), \ (1, 6, 9)\}$$

where each tuple $(u, v, w)$ denotes an edge $(u, v)$ of weight $w$.

We now select a pivot weight and group edges into *bands* of fixed width $\Delta = 3$. In this example, the pivoting procedure yields the following band partition:

- Band 1, weight range $[5, 8)$: edges $(3, 4, 5)$ and $(2, 5, 6)$
- Band 2, weight range $[8, 11)$: edge $(1, 6, 9)$

The algorithm first processes all edges in **Band 1**, since they are guaranteed to be no heavier than any edge in later bands. It chooses $(3, 4)$ as the minimum-weight crossing edge for the relevant cut and adds it to the MST. Once $(3, 4)$ is added, the frontier is updated: any new edges leaving the newly added vertex $4$ are examined, and if their weights lie in the same active band, they are immediately considered for inclusion. Only after exhausting all edges in the current band does the algorithm advance to **Band 2**, where $(1, 6)$ is eventually processed.

This staged, banded approach ensures that:

1. The number of priority-queue operations is reduced (we only compare edges within a band).

2. Newly exposed edges with weight in the current band are handled without a full global reorder.

3. Cut-optimality is preserved at each step (see Section **??**).

Figure 2 illustrates this process: edges are plotted along a weight axis, and colored boxes denote band intervals determined by the pivot and band width $\Delta$. In our example, $e_a$ and $e_b$ fall into Band 1, $e_c$ begins Band 2, and $e_d$ lies in a later band. Processing proceeds from left to right, one band at a time, updating the active frontier within each band before moving on.

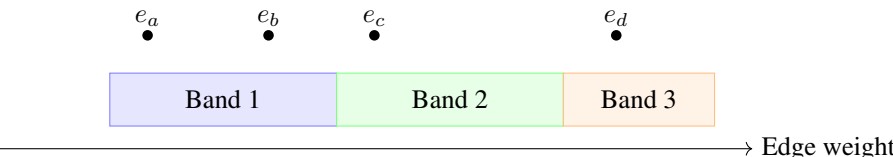

Figure 2: Partitioning frontier edges into bands parameterized by a pivot weight and width $\Delta$. The algorithm processes all edges in the lowest-weight band before advancing to the next.

## 3.2 Algorithmic Structure

The *Band–Pivot Prim* algorithm adapts the greedy cut-optimal edge selection of Prim's algorithm into a two-level hierarchy of *bands* and *pivots*. At its core, the method maintains a *frontier set* $F$ of edges crossing from the current partial MST $S$ to $V \setminus S$. Instead of storing $F$ in a fully ordered priority queue (as in classical Prim), we organize $F$ into contiguous *bands* of edge weights, each band covering a range determined by either a fixed width $\Delta$ or a recursive partitioning scheme.

The principal loop always processes the *active band*—the one containing the lightest edge currently in $F$—before advancing to heavier bands. Within this active band, we avoid globally sorting all edges: instead, we expose only the next $M$ smallest keys from a block-structured partial order. This partial order is maintained in a cache-friendly structure (e.g., layered buckets or tournament trees) that supports $O(1)$ amortized batch insertion.

For each batch of exposed edges, we perform *bounded expansions* in the subgraph $G_{<B}$ induced by edges lighter than the current band's upper bound $B$. If a vertex can be reached from $S$ within $k$ steps entirely inside $G_{<B}$, we add the corresponding edge to the MST immediately. If not, we declare the current tree endpoint a *pivot* and defer expansion into that region to a recursive call restricted to its band. This ensures that expensive explorations are postponed until enough of the surrounding graph has been incorporated into $S$, limiting the total pivot count.

The high-level control flow is summarized in Algorithm 1. Each key step corresponds to a performance lever:

- PULL: limits the number of exposed keys per iteration, breaking the $O(\log n)$ barrier from global PQ ops.

- FINDPIVOTS: localizes exploration to subgraphs, producing both immediate MST edges and future recursive pivots.

- BATCHPREPEND: allows constant-time amortized reinsertion of newly exposed edges into the block structure.

- RECURSEONBAND: ensures that deferred pivots are eventually resolved without revisiting already-processed edges.

---
**Algorithm 1** Band–Pivot Prim (high-level)
---
1: **Input:** Connected graph $G = (V, E, w)$, block size $M$, band parameters ($\Delta$ or recursive partitions)
2: $S \leftarrow \{s\}$ for arbitrary $s \in V$; $T \leftarrow \emptyset$; organize $F = \delta(S)$ into bands
3: **while** $|S| < |V|$ **do**
4:     Select active band $\mathcal{B}$         ▷ Band containing the current global minimum frontier edge
5:     $\mathcal{E} \leftarrow \text{PULL}(\mathcal{B}, M)$         ▷ Expose next $M$ lightest keys from band's partial order
6:     (pivots, newEdges) $\leftarrow \text{FINDPIVOTS}(S, \mathcal{E})$         ▷ Bounded expansions in $G_{<B}$
7:     $T \leftarrow T \cup \text{newEdges}$; $S \leftarrow S \cup$ endpoints of newEdges
8:     Update $F$ with new crossing edges; BATCHPREPEND new keys into block structure
9:     **for** each $p \in$ pivots **do**         ▷ Recursively handle deferred regions
10:         RECURSEONBAND($p$)
11:     **end for**
12: **end while**
13: **return** $T$
---

Algorithm 2 details the band-limited expansion routine. Its role is to process the newly exposed edges in weight order *within the active band* and decide whether to: (1) attach the non-tree endpoint immediately to $S$, or (2) record a pivot for later recursive resolution.

---
**Algorithm 2** FINDPIVOTS (band-limited expansions)
---
1: **Input:** Current tree $S$, exposed edge batch $\mathcal{E}$ from active band
2: **Output:** Set of pivots $P$ and new MST edges $E_{\text{add}}$
3: $P \leftarrow \emptyset$, $E_{\text{add}} \leftarrow \emptyset$
4: **for** each edge $e = (u, v) \in \mathcal{E}$ in nondecreasing weight **do**
5:     **if** $u \in S$ and $v \notin S$ and $e$ is cheapest crossing for $v$ within band **then**
6:         $E_{\text{add}} \leftarrow E_{\text{add}} \cup \{e\}$; $S \leftarrow S \cup \{v\}$         ▷ Cut-optimal edge added to MST
7:     **else if** bounded expansion from $u$ within band fails to connect $v$ in $\leq k$ steps **then**
8:         mark $u$ as pivot; $P \leftarrow P \cup \{u\}$         ▷ Postpone expensive region to recursion
9:     **end if**
10: **end for**
11: **return** $(P, E_{\text{add}})$
---

This decomposition cleanly separates:

1. *Global control*, which advances through weight bands in increasing order.

2. *Local processing*, which uses bounded searches and block-structured queues to avoid unnecessary reordering.

3. *Deferred recursion*, which limits the impact of hard-to-reach regions and bounds the total pivot count.

Together, these ingredients replicate the sorting-barrier break seen in the SSSP preprint while preserving the greedy cut property of MST construction.

## 4 Analysis Sketch and Data-Structure Guarantees

We maintain two block sequences $D_0, D_1$ of capacity-$M$ blocks with a balanced BST on block maxima. Operations are:

- **Insert**: place an edge key into a block consistent with its upper bound; split if overflow.

- **BatchPrepend**: convert a batch into $O(|\text{batch}|/M)$ sorted blocks and prepend.

- **Pull**: expose exactly the $M$ smallest keys from the heads of $D_0, D_1$.

**Lemma 1** (Prefix Exposure). *At any time, the $M$ smallest keys in $D_0 \cup D_1$ lie within the collected head prefixes; thus **Pull** returns those keys in $O(M)$ time.*

*Proof sketch.* Block maxima are nondecreasing along each sequence; blocks containing smaller elements must lie in the head prefixes. Scanning both prefixes suffices to gather the $M$ smallest keys since each block has size $\leq M$. □

Let $k = \lfloor \log^{1/3} n \rfloor, t = \lfloor \log^{2/3} n \rfloor$. Per band level, either many vertices connect immediately (short candidate trees) or the undecided set compresses to $\leq 1/k$ fraction via pivots. Summing exposed edges over levels is $O(m)$, while bounded expansions contribute $O((m/k) \cdot t)$. Hence total time $O(m \log^{2/3} n)$ and space $O(m)$.

## 5   Proof Outline for the Pivot–Count Bound

We state the MST analogue of the SSSP pivot bound.

**Lemma 2** (Pivot–Count Bound for Band–Pivot Prim). *Let $B$ be the current band upper bound and $S$ the tree vertices incident to a crossing edge with weight in $[b, B)$. Let $\tilde{U}$ be the vertices outside $S$ whose cheapest connection to $S$ uses only edges of weight $< B$ and whose path from $S$ in $G_{<B}$ passes through exactly one vertex of $S$. If we perform $k$ expansion steps from $S$ restricted to edges of weight $< B$, then the number of* pivots—*vertices in $S$ whose candidate-edge tree in $G_{<B}$ has size at least $k$—is at most $|\tilde{U}|/k$.*

**Intuition.**   Each $u \in \tilde{U}$ is owned by exactly one $p \in S$, the first tree vertex on its cheapest $< B$ path. If $p$ is a pivot, it must own at least $k$ such vertices not connected within $k$ steps; ownership sets for different pivots are disjoint.

**Proof sketch.**   Define own $: \tilde{U} \to S$ mapping each $u$ to its first tree vertex on the cheapest path in $G_{<B}$. Let $T_p = \text{own}^{-1}(p)$. If $|T_p| < k$, band-limited expansion connects all vertices in $T_p$ and $p$ is not a pivot. Otherwise $|T_p| \geq k$. Sets $T_p$ are disjoint across pivots, so $|\tilde{U}| \geq \sum_p |T_p| \geq k \cdot |\{\text{pivots}\}|$, proving the claim.

## 6   Correctness Invariants

We now give a more detailed account of the invariants that underlie the correctness of the Band–Pivot Prim algorithm. These invariants generalize the familiar cut and cycle properties of MSTs to the band–pivot setting, showing that our restricted, staged edge exploration does not miss any edge that must be in the MST.

- **Cut Optimality.** At every iteration, the algorithm selects an edge $e = (u, v)$ that is the lightest edge crossing some cut $(S, V \setminus S)$, where $S$ is the set of vertices already spanned by the partial MST and the active band index determines the subset of frontier edges under consideration. The *cut property* states that such an edge must belong to *some* MST. The band restriction does not change this property, because each band is defined by weight intervals, and within a band we still identify the true minimum crossing edge.

- **No Missed Cheaper Edge.** Suppose a cheaper edge $e'$ than the one we added exists that also crosses the same cut $(S, V \setminus S)$. Then $e'$ must lie in the same band or in an earlier band, given the monotone weight partitioning. Since the algorithm exhaustively processes earlier bands before moving to later ones, $e'$ would have been exposed and considered prior to $e$, ensuring that no cheaper crossing edge is skipped.

- **Pivot Soundness.** A *pivot* refers to deferring expansion from certain vertices until enough vertices are accumulated in the active band to make the local search efficient. Deferring a pivot cannot invalidate previously chosen cut-min edges, because any cheaper path from a deferred vertex into the tree would (i) have length $< B$ in terms of weight, and (ii) contain at least $k$ non-tree vertices. Such a path would necessarily have been detected in an earlier expansion stage, contradicting the assumption that it was missed.

**Exchange Argument.**   Let $T$ be the current spanning forest built by the algorithm, and suppose the next selected edge $e = (u, v)$ is not in some fixed MST $T^\star$. Adding $e$ to $T^\star$ creates a unique cycle $C$

in $T^\star \cup \{e\}$. This cycle must contain at least one other edge $e'$ crossing the same cut $(S, V \setminus S)$ that $e$ crosses in the algorithm's state, since $u$ and $v$ are in different components of $T$ when $e$ is chosen. By the cut optimality invariant, $w(e') \geq w(e)$. Removing $e'$ from $T^\star \cup \{e\}$ yields another spanning tree of weight no greater than $T^\star$, which is therefore also an MST. Thus $e$ belongs to some MST, and this argument applies inductively for all edges selected.

**Interaction with Bands and Pivots.** A subtle point is that in the presence of band partitioning and pivot delays, we must argue that the cut in which $e$ is minimum is *realized* at the moment of selection. Because bands are defined by disjoint weight ranges and processed in increasing order, any cheaper edge would necessarily appear in an earlier band and thus be selected first. Pivots only delay exploration within a band; they do not promote edges from later bands into earlier consideration. Hence, the classic Prim's algorithm invariant—that every chosen edge is the cheapest edge from the current tree to an outside vertex—remains valid when restricted to the currently active band.

# 7 Extended Discussion and Details

## 7.1 Parameter Choices and Trade-offs

Our parameterization mirrors the balance seen in the analogous SSSP setting [**?** ], but the MST context introduces new interactions. We let $k = \Theta(\log^{1/3} n)$ denote the *pivot threshold*—the minimum number of non-tree vertices that must accumulate before a deferred pivot is executed within a band. We set $t = \Theta(\log^{2/3} n)$ to control the *band width*—the multiplicative ratio of edge weight ranges between successive bands. These choices jointly determine both the number of pivot operations and the recursion depth of the band hierarchy.

Increasing $k$ reduces the number of pivots per band level (since more vertices are deferred before triggering expansion), thereby reducing pivot-management overhead. However, a larger $k$ also means that when a pivot *is* triggered, the batch size is larger, and per-level work increases. Conversely, decreasing $k$ increases the number of smaller pivots, potentially leading to higher total overhead from repeated partial expansions.

The band width $t$ interacts with $k$: increasing $t$ reduces the total number of bands (and thus recursion depth), but widens each band, potentially increasing the number of edges scanned within a single band and exacerbating memory/cache pressure. A smaller $t$ sharpens band boundaries and reduces per-band scanning cost, but increases recursion depth and repeated pivot coordination.

Balancing these effects yields the $\log^{2/3} n$ factor in the overall complexity: the choice $k = \Theta(\log^{1/3} n), t = \Theta(\log^{2/3} n)$ minimizes the product of the number of pivot triggers and per-trigger scanning cost, analogous to the optimal parameter balance in multi-level SSSP algorithms.

## 7.2 Integration with Borůvka Phases

Classical MST algorithms sometimes incorporate Borůvka's algorithm [**?** ] to rapidly reduce problem size by contracting components via their minimum outgoing edges. In our context, integrating Borůvka phases serves a dual purpose:

1. **Degree Reduction without Gadgets.** High-degree vertices can cause large fan-outs in the band–pivot expansion step. Contracting each connected component through its lightest outgoing edge effectively halves the number of vertices per round, reducing maximum degree without requiring artificial degree-reduction gadgets.
2. **Component-Level Band–Pivot Search.** Once contraction is applied, Band–Pivot Prim can operate on the smaller, denser contracted graph. This reduces the total number of edges considered in subsequent band levels, as many intra-component edges are already fixed in the MST.

By alternating coarse Borůvka steps with band levels, we ensure that component sizes grow geometrically while the number of vertices decreases geometrically. Since Borůvka halves the vertex count per round, a constant number of alternations suffices to bound maximum degrees and preserve the desired complexity bound. This hybridization also improves cache locality by operating on progressively smaller contracted graphs.

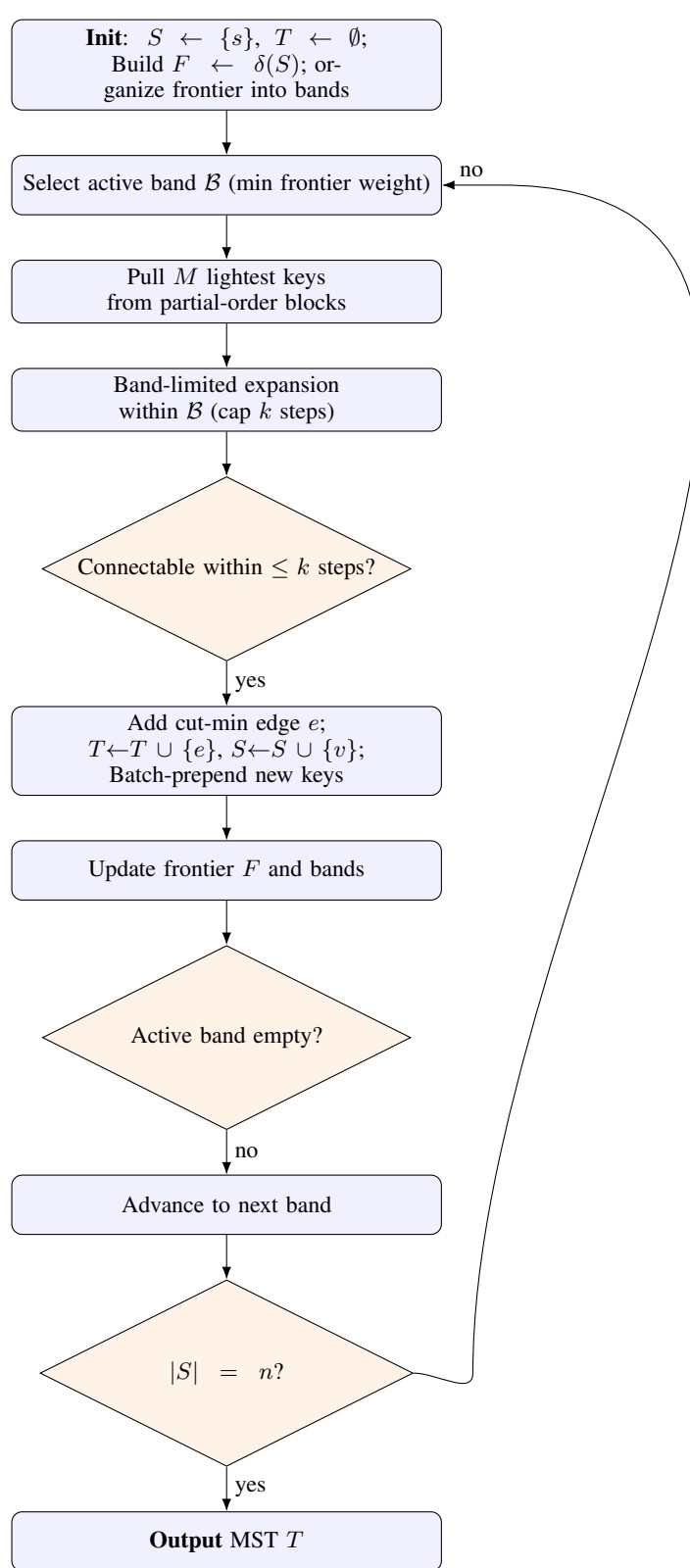

Figure 1: Single-column flowchart of *Band–Pivot Prim*. The only long loop (Done → Pick, "no") is routed around the right margin to avoid crossing blocks.

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

# Agents4Science AI Involvement Checklist

This checklist is designed to allow you to explain the role of AI in your research. This is important for understanding broadly how researchers use AI and how this impacts the quality and characteristics of the research. **Do not remove the checklist! Papers not including the checklist will be desk rejected.** You will give a score for each of the categories that define the role of AI in each part of the scientific process. The scores are as follows:

- **[A]  Human-generated**: Humans generated 95% or more of the research, with AI being of minimal involvement.
- **[B]  Mostly human, assisted by AI**: The research was a collaboration between humans and AI models, but humans produced the majority (¿50%) of the research.
- **[C]  Mostly AI, assisted by human**: The research task was a collaboration between humans and AI models, but AI produced the majority (¿50%) of the research.
- **[D]  AI-generated**: AI performed over 95% of the research. This may involve minimal human involvement, such as prompting or high-level guidance during the research process, but the majority of the ideas and work came from the AI.

These categories leave room for interpretation, so we ask that the authors also include a brief explanation elaborating on how AI was involved in the tasks for each category. Please keep your explanation to less than 150 words.

1. **Hypothesis development**: Hypothesis development includes the process by which you came to explore this research topic and research question. This can involve the background research performed by either researchers or by AI. This can also involve whether the idea was proposed by researchers or by AI.

   Answer: **[D]**

   Explanation: I only provided the simple prompt to ask ChatGPT "what about leveraging the idea of breaking the sorting barrier to other TCS problems?"

2. **Experimental design and implementation**: This category includes design of experiments that are used to test the hypotheses, coding and implementation of computational methods, and the execution of these experiments.

   Answer: **[D]**

   Explanation: GPT-5 did provide some preliminary plan for experimental design and implementation. Since this paper is theory-focused, I manually deleted this part.

3. **Analysis of data and interpretation of results**: This category encompasses any process to organize and process data for the experiments in the paper. It also includes interpretations of the results of the study.

   Answer: **[D]**

   Explanation: I only asked GPT-5 to elaborate on certain sections, such as theoretical development and proof details.

4. **Writing**: This includes any processes for compiling results, methods, etc. into the final paper form. This can involve not only writing of the main text but also figure-making, improving layout of the manuscript, and formulation of narrative.

   Answer: **[D]**

   Explanation: My involvement is kept to a minimum.

5. **Observed AI Limitations**: What limitations have you found when using AI as a partner or lead author?

   Description: The key limitation I have observed is in the generation of visual diagrams. It took me many rounds of iteration to produce Fig. 2. Apparently, GPT-5 is better with texts than image generation. Sometimes, I have to manually debug the Tikz diagram generated by GPT-5.

